# A Case of Clinical Uncertainty Solved: Giant Cell Arteritis with Polymyalgia Rheumatica Swiftly Diagnosed with Long Axial Field of View PET

**DOI:** 10.3390/diagnostics12112694

**Published:** 2022-11-04

**Authors:** Pieter H. Nienhuis, Joyce van Sluis, Johannes H. van Snick, Andor W. J. M. Glaudemans, Sofie Meijering, Elisabeth Brouwer, Riemer H. J. A. Slart

**Affiliations:** 1University of Groningen, University Medical Center Groningen, Medical Imaging Center, Department of Nuclear Medicine and Molecular Imaging, 9713 GZ Groningen, The Netherlands; 2Ommelander Hospital Groningen, Department of Internal Medicine, 9679 BJ Scheemda, The Netherlands; 3University of Groningen, University Medical Center Groningen, Department of Rheumatology and Clinical Immunology, 9713 GZ Groningen, The Netherlands; 4University of Twente, Faculty of Science and Technology, Biomedical Photonic Imaging Group, 7522 NB Enschede, The Netherlands

**Keywords:** LAFOV PET/CT, 18F FDG PET/CT, giant cell arteritis, polymyalgia rheumatica, vasculitis

## Abstract

The clinical presentation of giant cell arteritis (GCA) is often nonspecific. Differentiating GCA from infectious, malignant, or other autoimmune pathology based on signs, symptoms, and laboratory parameters may therefore be difficult. Fluorine-18-fluorodeoxyglucose (^18^F-FDG) positron emission tomography/computed tomography (PET/CT) imaging is an established tool in the diagnostic workup of GCA. An advantage of ^18^F-FDG-PET/CT is its ability to assist in the differential diagnosis by being able to demonstrate infection, inflammation, and malignancy when used in conjunction with clinical and laboratory data. Downsides to the use of ^18^F-FDG-PET/CT include its relatively low spatial resolution, associated radiation exposure, and the relatively long duration of imaging, causing limited availability and patient inconvenience. The advent of long axial field-of-view (LAFOV) PET/CT systems allows for PET imaging at a reduced imaging time or reduced tracer dose while maintaining high image quality. Here, we provide the first reported case of a patient with GCA and polymyalgia rheumatica (PMR) diagnosed using LAFOV PET/CT imaging. The patient presented in this case report had already been experiencing nonspecific symptoms for several years for which no cause was found. Lab investigations showed increased inflammatory parameters as well as persistent anemia. ^18^F-FDG LAFOV PET/CT attained high-quality images with clear signs of GCA and PMR even at 1 min of scan duration.

## 1. Introduction

Giant cell arteritis (GCA) is a large vessel vasculitis that commonly affects the aorta and its major branches (large vessel GCA) and can also affect the medium-sized arteries in the head and neck (cranial GCA). It is the most frequently diagnosed primary vasculitis in the United States and belongs to the same disease spectrum as polymyalgia rheumatica (PMR) [1]. Almost half of the patients with GCA have PMR, and 20% of PMR patients develop GCA [2].

Diagnosing GCA is difficult because signs, symptoms, and laboratory investigations are often nonspecific [3]. Especially patients with large vessel GCA may solely present with nonspecific systemic symptoms, such as malaise, fever, and weight loss [4].

Fluorine-18-fluorodeoxyglucose (^18^F-FDG) positron emission tomography/computed tomography (PET/CT) is a diagnostic imaging technique with a high accuracy for diagnosing large vessel GCA [5]. In part due to its whole-body nature, ^18^F-FDG-PET/CT can detect inflammation in all the medium- and large-sized vessels throughout the body and is able to differentiate between different vasculitis types by pattern recognition. Likewise, ^18^F-FDG-PET/CT can be of significant value in confirming or discarding a diagnosis of PMR [6]. It shows great concordance with a clinical diagnosis of PMR and can indicate the extent of disease for pure PMR, as well as for the entire spectrum of disease [7,8]. Importantly, other infectious, inflammatory, or malignant processes may be detected by ^18^F-FDG-PET/CT.

Limitations to the use of conventional ^18^F-FDG-PET/CT imaging include the exposure to radiation, the limited spatial resolution of 4–5 mm, relatively low sensitivity for low grade inflammatory diseases, the relatively long duration of imaging (10 to 30 min), and the associated limited availability and patient inconvenience [9].

Several new PET/CT systems with a long axial field of view (LAFOV) have become available in recent years [10,11]. An extended axial field of view results in a substantial increase in sensitivity, which allows for improved image quality, even at shorter scan durations and/or lower injected dose [12] compared with conventional PET/CT systems.

Recently, a LAFOV Biograph Vision Quadra PET/CT system [13] was installed at the department of Nuclear Medicine and Molecular Imaging at the University Medical Center Groningen. The Biograph Vision Quadra PET/CT system with an axial field of view of 106 cm essentially consists of four interconnected ‘digital’ Biograph Vision PET systems [14] equipped with silicon-based photomultiplier detector elements.

LAFOV PET/CT imaging has the potential to improve imaging of inflammatory conditions, such as GCA and PMR [13]. Here, we report of the first case of GCA and PMR imaged on a LAFOV PET/CT system. We subsequently discuss our findings and relate this to future perspectives for clinical practice and research.

## 2. Case Presentation

The case concerns a 55-year-old female who was referred to the outpatient clinic of the internal medicine department at the Ommelander Hospital Groningen with persistent mild anemia (hemoglobin 6.7 mmol/L, normal 7.5–10 mmol/L) in the past three months and slightly elevated ferritin (241 µg/L, normal 15–130 µg/L). Her medical history revealed that she had been experiencing pain in the left knee and hands for the past two years. Additionally, she recently visited a neurologist because of a misty feeling in her head and a sensation of falling to the right for which she would be undergoing further investigations.

At the time of presentation at the internal medicine’s outpatient clinic, her main complaint was a sensation of fatigue in her legs after exertion. Further history taking revealed that she had lost 12 kg of weight in the past year while actively pursuing an active lifestyle and healthy eating. Additionally, she noticed that lately, she was tired earlier in the evenings. She reported no fever, night sweats, or stiffness/pain around the shoulder or hip girdle. History taking was also negative for typical symptoms of cranial GCA with no headache, jaw claudication, or visual symptoms. In addition, no abnormalities, including no limited range of movement, were noted during the physical exam.

In addition to persistent normocytic anemia and slightly elevated ferritin, she had an increased c-reactive protein (CRP) of 81 mmol/L, and high immunoglobulins A and G (4.4 and 20.0 g/L, respectively).

A wide differential diagnosis included large vessel vasculitis, (hematologic) malignancy, and rheumatic diseases. An ^18^F-FDG-PET/CT scan was performed to help differentiate between these potential causes. The indication for the ^18^F-FDG-PET/CT scan was stated as “anemia of inflammation and increased CRP without specific signs or symptoms”.

The patient was instructed to fast for 6 h and drink 1 L of water before the intravenous ^18^F-FDG injection. Plasma glucose levels were 4.4 mmol/L before activity administration. Patient weight was 78 kg; therefore, she received a standard weight-based (3 MBq/kg) intravenous injection of 230 MBq ^18^F-FDG activity followed by a whole body 10 min listmode PET acquisition at 60 min post injection. PET data were acquired using a single static bed position covering 106 cm (approximately from vertex to mid-thigh). PET data acquired for 10 min were reconstructed and, in addition, image reconstructions including 3 min and 1 min of the acquisition time were obtained. Images were reconstructed using the vendor-recommended clinically optimized protocol for optimal image quality, consisting of 3D OSEM with four iterations, five subsets, a matrix size of 440 × 440 × 708 with a voxel size of 1.6 × 1.6 × 1.5 mm^3^, time-of-flight application, resolution modeling, and no filtering. Data were acquired using a maximum ring difference of 85.

^18^F-FDG-PET/CT showed physiologic uptake in the brain, salivary glands, mediastinum, liver, spleen, kidneys, and bladder. Pathologic uptake was seen at the walls of the aorta, brachiocephalic trunk, common carotid arteries, and subclavian arteries (Figure 1). No other abnormal uptake was seen—neither in the chest and neck—nor did a low-dose CT show any infiltrate, ground glass, or nodular pathology. The pathologic uptake surrounding the aorta continued into the abdominal aorta and the common iliac arteries, in combination with a pathologic FDG uptake surrounding the sternoclavicular joints, acromioclavicular joints, shoulder joints, greater trochanters, and spinous processes; this is highly suggestive of GCA with PMR. 

A diagnosis of GCA and PMR was made based on the ^18^F-FDG-PET/CT interpretation as described above, in combination with the clinical presentation and increased CRP. After three days of treatment with high-dose prednisolone (60 mg/day), her symptoms reduced drastically. The patient noticed an increase of symptoms during prednisolone tapering, which was managed by increasing the dose. She is currently in follow-up, receiving a combination therapy of prednisolone and methotrexate.

## 3. Discussion

This case represents a category of patients in whom signs, symptoms, and initial lab investigations do not help to differentiate between malignancy, infections, and autoimmune diseases. The persistently increased CRP and anemia as demonstrated in this patient may fit to chronic inflammation, but also to malignancy. 

When encountered with such a wide differential diagnosis ranging from inflammatory disorders to malignancy, ^18^F-FDG-PET/CT may open new avenues in the diagnostic process. ^18^F-FDG-PET/CT is particularly useful because of its proven value in many applications. This encompasses especially malignancies, but also infection and inflammation [9,15]. Historically, the applicability of PET was limited because of its low spatial resolution and sensitivity [16]. This significantly improved with the advent of newer generation (digital) PET/CT systems [14]. For example, only recently have newer generation PET systems been able to visualize inflammation of the cranial arteries in cranial GCA [17,18,19].

With the advent of LAFOV PET, even smaller anatomical details may be identified while noise is significantly reduced. Hence, a low-grade uptake may be identified, further increasing sensitivity and diagnostic applicability. Moreover, specificity is also increased because multiple patterns of FDG uptake may be distinguished. This allows for differentiation between atherosclerosis and vasculitis, as well as between different types of vasculitis [20]. In relation to the presented patient case, higher specificity increases the use of ^18^F-FDG-PET/CT to aid the diagnostic process in cases of clinical uncertainty by more accurate identification of pathological FDG uptake. 

Additionally, high-image quality may still be achieved while reducing the injected tracer dose or scan duration, even after 1 min of acquisition time. Coincidentally, short-imaging protocols increase quality by reducing motion artifacts [21]. More importantly, it may also significantly increase patient friendliness, further improving accessibility to ^18^F-FDG-PET/CT imaging. Higher accessibility for the general patient population may also be due to increased patient throughput. Alternatively, higher accessibility for radiosensitive populations, such as children and pregnant women, may be achieved by a reduction in injected tracer dose [22].

Compared to a short axial field-of-view PET system, LAFOV PET allows for markedly improved lesion quantification [23]. More accurate uptake quantification may aid the standardization and further improve the reproducibility of imaging findings [13]. Currently, no semi-quantitative assessment methods are recommended for clinical use in GCA or PMR [24]. In GCA, many different semi-quantitative assessment methods are used in the current literature. Hence, the available data are too heterogeneous to adequately compare them to visual scoring methods [25]. Improving reproducibility with LAFOV PET may help identify reliable semi-quantitative parameters for standardized use in future studies, which may, in turn, be measured against the current standard visual scoring methods.

Lastly, LAFOV PET opens the possibility to reduce the injected dose of radionuclides with longer half-lives, thereby reducing the risk associated with increased radiation exposure from such radionuclides. This may enable PET imaging with monoclonal antibodies in patients with inflammatory conditions, such as GCA and PMR [13,26]. Radiotracers that target specific immune cells hold great promise to accurately determine not only the presence of inflammation, but also the nature of the cells involved and the effectiveness of treatment.

## 4. Conclusions

The first case of GCA and PMR diagnosed using LAFOV PET is described. Despite clear clinical uncertainty regarding the diagnosis, high-quality images obtained by LAFOV PET provided clear confirmation for a diagnosis of GCA and PMR while having a significantly reduced scan duration.

## Figures and Tables

**Figure 1 diagnostics-12-02694-f001:**
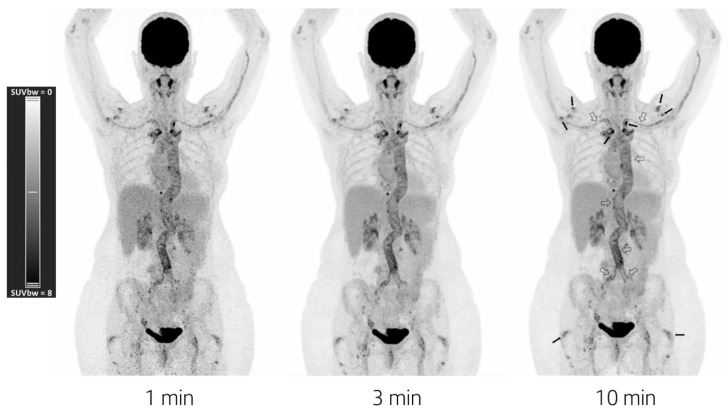
Maximum intensity projection PET images acquired for 1 min (**left**), 3 min (**middle**), and 10 min (**right**). Images were reconstructed using a vendor-recommended clinically optimized protocol for optimal image quality, consisting of 3D OSEM with 4 iterations, 5 subsets, a matrix size of 440 × 440 × 708 with a voxel size of 1.6 × 1.6 × 1.5 mm^3^, time-of-flight application, resolution modelling, and no filtering. Data were acquired using a maximum ring difference of 85. In all three images, ranging from very fast to standard acquisition time, one can see the following typical features of GCA and PMR: increased uptake in the aortic wall; the common iliac arteries; and the subclavian and axillary arteries due to GCA activity (hollow arrows in right image), and moderately to highly increased uptake around the shoulders, sternoclavicular joints, acromioclavicular joints, and greater trochanters due to PMR activity (filled in black arrows right image).

## Data Availability

Not applicable.

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
