# Peer review of "A Case of Clinical Uncertainty Solved: Giant Cell Arteritis with Polymyalgia Rheumatica Swiftly Diagnosed with Long Axial Field of View PET"

_diagnostics, 2022, doi:10.3390/diagnostics12112694_

Round 1

Reviewer 1 Report

This manuscript provides a summation of an interesting diagnosis of giant cell arteritis with polymyalgia rheumatica specifically by use of long axial field of view in positron emission tomography/ computed tomography (PET/CT). The report is clearly and succinctly written with a good description of the background to the case and the rationale for the imaging approach taken. The imaging methods used in the diagnosis will clearly be of interest to specialists in the field and the report expands our understanding of the capabilities of this form of PET/CT imaging. There are some minor points that the authors could address and these are detailed below.

1.       Whilst an extensive literature review is not expected in this succinct paper format, there are some recent references that are particularly pertinent to this topic that have not been included in the references list but I feel are worth citing. For example: Emamifar et al. “The Utility of 18F-FDG PET/CT in Patients with Clinical Suspicion of Polymyalgia Rheumatica and Giant Cell Arteritis: A Prospective, Observational, and Cross-sectional Study” ACR Open Rheumatol 2020 2(8) 478-490; Rehak et al. “18F-FDG PET/CT in Polymyalgia Rheumatica – A Pictorial Review” Br J Radiol 2017 90(1076) 20170198.

2.       Page 2 line 79 “. . . a review of systems.” I presume you mean “. . . a review of symptoms.”

3.       Page 4 lines 149-150 “Additionally, high image . . . maximum of 1 minute.” In Figure 1 the shortest scan duration shown is one minute but here in the text you are talking about a number of scans of different duration up to a maximum of one minute. Do you actually mean reduction of scan duration down to a minimum of one minute? Clarify your meaning in the manuscript.

4.       Page 4 lines 154-155 “. . . which complements the . . . reduced tracer dose.” Can you offer any comment on the degree to which this tracer dosage might be able to be reduced based on the findings from this case study?

5.       Page 4 lines 174-175 “. . . high quality images . . . diagnosis of GCA and PMR . . .” Whilst you have provided a good description of the case and the capabilities of the PET/CT imaging system, I think that the link between the image results obtained and the diagnostic criteria used could be made stronger and clearer in the text. In other words, can you be more explicit about which features in the imaging you used to make your final diagnosis, i.e. an expansion of what you have written at lines 112-113? This would be particularly useful for readers who are not so familiar with this field of imaging.

Author Response

Please see the attachment. Points raised are cursive and replied to under each comment.

Reviewer 2 Report

It is an interesting case for giant cell arteritis with polymyalgia rheumatica.

1. Keywords (line 31): deleted GCA and PMR. [18F] change into 18F

2. line 69: 6.7 mmol/L add Hgb ? and normal reference, ferritin add the normal reference.

3. line 147: 18F changes into 18F.

4. References recheck, please.

Author Response

(The authors gave the same response as above.)

Reviewer 3 Report

I congratulate with the authors for the very interesting case report.

Some minor considerations:

-          In the introduction, before specifying that in authors’ department QUADRA has been installed, I would suggest to add a brief paragraph to explain what LAFOV PET/CT is and that there are also other devices than QUADRA.

-          In the Discussion, the authors write: “high image quality may still be achieved while reducing the injected  tracer dose or scan duration, even up to a maximum of 1 minute. Coincidentally, short imaging protocols increase quality by reducing motion artifacts.” I would stress this important topic (fast /very fast protocols) by adding the following  reference “Eur J Nucl Med Mol Imaging. 2022 Aug;49(10):3322-3327. doi: 10.1007/s00259-022-05791-z”

-          Figure is very illustrative. However, I would suggest to slightly change the legend (“In all three images, increased uptake is noticed in the aortic wall, the common iliac arteries, and the subclavian andaxillary arteries due to GCA activity”) to stress, for NM less-experts readers, that the images, although simulating different time-duration protocols (i.e. from very fast to standard), are of substantially equal quality.

Author Response

(The authors gave the same response as above.)
